# Differentiating Benign from Malignant Thyroid Tumors by Kinase Activity Profiling and Dabrafenib BRAF V600E Targeting

**DOI:** 10.3390/cancers15184477

**Published:** 2023-09-08

**Authors:** Riet Hilhorst, Adrienne van den Berg, Piet Boender, Tom van Wezel, Tim Kievits, Rik de Wijn, Rob Ruijtenbeek, Willem E. Corver, Hans Morreau

**Affiliations:** 1PamGene International BV, 5211 HH ‘s-Hertogenbosch, The Netherlands; rhilhorst@pamgene.com (R.H.);; 2Leiden University Medical Center, 2333 ZA Leiden, The Netherlandsj.morreau@lumc.nl (H.M.)

**Keywords:** thyroid, *BRAF* mutation, dabrafenib, kinase activity profiling, peptide microarray

## Abstract

**Simple Summary:**

Recurrent non-medullary thyroid cancer (NMTC) is difficult to treat and therapy options are limited. Of the available compounds, serine/threonine kinase (STK) inhibitors are currently widely used. However, this form of targeted therapy is not always effective and additional response-related biological information may improve both accuracy and efficacy. Using in-depth STK activity profiling, in this study, we aimed to determine whether benign and malignant thyroid tumors can be differentiated based on these profiles. In addition, we analyzed the impact of the BRAF V600E-specific inhibitor dabrafenib, as well as the generic RAF inhibitors sorafenib and regorafenib, in a subgroup of BRAF V600E and non-BRAF V600E papillary thyroid carcinomas. We demonstrate that STK activity profiling can differentiate benign from malignant thyroid tumors. Furthermore, the BRAF V600E-specific inhibitor dabrafenib can distinguish BRAF V600E from non-BRAF V600E papillary thyroid carcinomas. We conclude that STK activity profiling is beneficial when the goal is to differentiate benign from malignant thyroid tumors. In addition, this approach aids in the selection of likely effective (novel) kinase inhibitors for treatment of recurrent thyroid and other cancers.

**Abstract:**

Differentiated non-medullary thyroid cancer (NMTC) can be effectively treated by surgery followed by radioactive iodide therapy. However, a small subset of patients shows recurrence due to a loss of iodide transport, a phenotype frequently associated with BRAF V600E mutations. In theory, this should enable the use of existing targeted therapies specifically designed for BRAF V600E mutations. However, in practice, generic or specific drugs aimed at molecular targets identified by next generation sequencing (NGS) are not always beneficial. Detailed kinase profiling may provide additional information to help improve therapy success rates. In this study, we therefore investigated whether serine/threonine kinase (STK) activity profiling can accurately classify benign thyroid lesions and NMTC. We also determined whether dabrafenib (BRAF V600E-specific inhibitor), as well as sorafenib and regorafenib (RAF inhibitors), can differentiate BRAF V600E from non-BRAF V600E thyroid tumors. Using 21 benign and 34 malignant frozen thyroid tumor samples, we analyzed serine/threonine kinase activity using PamChip^®^peptide microarrays. An STK kinase activity classifier successfully differentiated malignant (26/34; 76%) from benign tumors (16/21; 76%). Of the kinases analyzed, PKC (theta) and PKD1 in particular, showed differential activity in benign and malignant tumors, while oncocytic neoplasia or Graves’ disease contributed to erroneous classifications. Ex vivo BRAF V600E-specific dabrafenib kinase inhibition identified 6/92 analyzed peptides, capable of differentiating BRAF V600E-mutant from non-BRAF V600E papillary thyroid cancers (PTCs), an effect not seen with the generic inhibitors sorafenib and regorafenib. In conclusion, STK activity profiling differentiates benign from malignant thyroid tumors and generates unbiased hypotheses regarding differentially active kinases. This approach can serve as a model to select novel kinase inhibitors based on tissue analysis of recurrent thyroid and other cancers.

## 1. Introduction

Non-medullary thyroid cancer (NMTC) accounts for 97% of all thyroid cancers and is the most common endocrine cancer worldwide. NMTC is currently the ninth most frequent cancer worldwide, with over a half million new cases annually, notably in Asia (GLOBOCAN 2020: http://globocan.iarc.fr/Pages/fact sheets population.aspx (accessed on 19 January 2023)).

Several types of thyroid lesions can be distinguished using routine microscopic examination. Benign thyroid lesions are typically classified as hyperplasia (HP), follicular adenoma (FA) or oncocytic adenoma (OA) [1,2]. Among the NMTCs, papillary thyroid cancer (PTC) and follicular thyroid cancer (FTC) are the most common. PTC and FTC comprise approximately 81% and 12% of all NMTC, respectively, which include variants of PTC. Oncocytic carcinoma of the thyroid (OCA) [2] (previously known as Hürthle cell carcinoma (HCC) or oncocytic variant of follicular thyroid carcinoma (FTC-OV)) [3,4] comprises around 3–4% of all NMTC and has been recognized by the WHO as a distinct subvariant. Anaplastic thyroid carcinoma (ATC) is the most aggressive form but accounts for less than 1% of all NMTC [5,6]. Well-differentiated NMTC can be curatively treated by surgery, with or without adjuvant radioactive iodide therapy. However, a subset of patients shows recurrence due to a loss of sodium iodide transport [7], with two forms primarily seen: OCA and PTC with the *BRAF* V600E variant.

Recurrent OCA is mainly characterized by a near-homozygous genome [3] and diverse DNA variants with or without mitochondrial DNA mutations [4,8,9], while recurrent PTC is typically associated with somatic *BRAF* V600E mutations (with increased frequencies of *hTERT* alterations) or gene fusions involving *ALK* or *NTRK*. The involvement of individual signaling pathways in recurrent iodide refractory thyroid cancer cases increases therapeutic options, such as the increasingly used VEGF pathway inhibitors. Another example is the serine/threonine protein kinase BRAF, a component of the MAPK signaling pathway [10]. Several drugs have been developed that interact with the serine/threonine kinase activity of BRAF or the BRAF V600E variant. The multi-kinase inhibitor sorafenib has been proven to be beneficial in the treatment of recurrent NMTC, despite its low affinity for BRAF V600E [11,12]. Furthermore, vemurafenib and dabrafenib were specifically designed to target this mutant and are effective against BRAF V600E-positive thyroid cancer [13,14]. Both inhibitors stimulate radioactive iodide uptake [15,16], and the results of a phase I trial in *BRAF* V600E-positive cancers have been published recently [17,18,19].

Next generation sequencing (NGS) revolutionized the detection of genetic variants suitable for targeted therapy, and has become an indispensable tool in daily diagnostic settings. While targeted therapy is often beneficial, it is also clear that is it is not a panacea.

Additional tools to better characterize tumors and improve understanding of underlying molecular biology might also be helpful. We propose that serine/threonine kinase (STK) activity profiling could be a rich source of biological information.

The aim of this study was therefore two-fold. First, we asked whether benign thyroid abnormalities and NMTC can be classified based on STK activity profiles, and whether these profiles can provide new insights concerning the differential activity of kinases between groups. Second, we investigated whether the BRAF V600E-specific inhibitor dabrafenib, as well as the RAF inhibitors sorafenib and regorafenib, induce differences in kinase inhibition between BRAF V600E and non-BRAF V600E thyroid tumors, and between recurrent and non-recurrent tumors. In order to address the first question, we used peptide microarrays to measure STK activity in lysates from 55 primary clinical tissue samples from benign and malignant thyroid tumors [20]. We demonstrate that STK activity profiling correctly differentiates benign and malignant thyroid tumors in 76% of cases. We also identified putative upstream kinases that are differentially active in benign and malignant tumors. Ex vivo spiking of PTC thyroid tumor lysates with the specific kinase inhibitor dabrafenib achieved partial differential inhibition of somatic *BRAF* V600E-positive versus non-*BRAF* V600E thyroid tumors and of recurrent versus non-recurrent tumors, which was not the case with regorafenib and sorafenib.

## 2. Materials and Methods

### 2.1. Patient Material

Patient material was processed immediately after surgery. A piece of fresh tumor was cut in half, one part was snap frozen and the second part was formalin-fixed and paraffin-embedded (FFPE). From the FFPE blocks and the fresh frozen tissue, 4 µm sections were cut and hematoxylin/eosin stained. An experienced pathologist (HM) performed morphological evaluation on both FFPE and fresh frozen tissue sections (see Appendix A for histology). All samples were processed in concurrence with the medical ethical guidelines as described in the Code Proper Secondary Use of Human Tissue (Dutch Federation of Medical Sciences, www.federa.org (accessed on 28 November 2022)). That Code has a system of ‘opt-out’ for further use in scientific research of coded human tissue, unless there are special circumstances. The current study, including the used ‘opt-out’ policy, was approved by the Medical Ethical Committee of the Leiden University Medical Centre, protocol no. B16.012.

### 2.2. Nucleic Acid Isolation, DNA Variant, and Fusion Analysis

Isolation of total nucleic acids was performed as described earlier [21]. In short, total nucleic acid (DNA and RNA) was isolated from 0.6 mm FFPE tissue cores (tissue microarrayer, Estigen OÜ, Tartu, Estonia) using a fully automated extraction procedure [22]. DNA variant analysis (e.g., *BRAF*, *NRAS*, *HRAS*, and *KRAS*) was performed using either a customized AmpliSeq Cancer Hotspot Panel or with Sanger sequencing, depending on the time period as previously described [3,23]. Additional *TERT* promoter variant (NM_198253.2; c.-57A>C, c.-124C>T, and c.-146C>T) analysis was performed by Sanger sequencing. The 8 non-tested HP cases did not significantly differ based on age of onset, gender, histological subtype, and genetic alterations distribution from 47 tested cases (Appendix A). Gene fusion analysis was performed as described earlier [21] on selected DNA-variant negative cases. Data analysis was performed using the online Archer Analysis software version 5.0 (http://analysis.archerdx.com (accessed on 4 September 2023)). Only ‘strong-evidence’ fusions within the software annotation were reported. Furthermore, *BRAF*/*RAS* point mutations were reported based on DNA/RNA reads. The total number of reads and the fractions of unique reads/RNA reads were documented for all samples as possible quality indicators.

Allelic state analysis, providing chromosomal copy number and loss of heterozygosity information, was performed as described earlier [24].

### 2.3. Materials

Mammalian Protein Extraction Reagent (M-PER^TM^, Cat. No. 78503), Halt^TM^ protease inhibitor cocktail EDTA free (Cat. No. 78437), and Halt^TM^ phosphatase inhibitor cocktail (Cat. No. 78420) were from ThermoFisher Scientific (Waltham, MA, USA). Dabrafenib (GSK2118436, Cat. No. S2807), regorafenib hydrochloride (Cat. No. S4947), and sorafenib tosylate (BAY 43-9006, Cat. No. S1040) were from SelleckChem (Houston, TX, USA). Stock solutions were prepared in 100% DMSO and stored at −20 °C prior to use.

Protein serine/threonine kinase peptide microarrays and reagents were obtained from PamGene International BV (‘s-Hertogenbosch, The Netherlands). The PamChip^®^ peptide microarrays used for this study contained either 240 or 140 substrate peptides. Experiments were performed with 96-array plates on a PamStation 96^TM^.

### 2.4. Preparation of Lysates

Cryo-sections of 10 µm thickness were cut (Leica Frigocut, 2800E, Wetzlar, Germany), collected in a pre-cooled (−20 °C) polypropylene tube, and stored at −80 °C until further use.

The 10 µm sections were lysed in M-PER lysis buffer (100 µL in the first lysis batch (55 samples), 20 µL in second and third lysis batches (PCTs only)), supplemented with protease and phosphatase inhibitor cocktail (1/100 diluted) as described by Hilhorst et al. [20]. The supernatants were aliquoted and stored at −80 °C until use. For every experiment, a new aliquot was used. The protein content of the lysates was determined with the Bradford protein assay (Bio-Rad, Hercules, CA, USA).

### 2.5. Kinase Activity Assays

For 55 thyroid samples, kinase activity was determined at 30 °C with 10 µL of lysate in the presence of both primary and secondary antibodies on 96-array plates in the presence or absence of 400 µM ATP (Sigma, Cat. No. A2383, St. Louis, MO, USA) on a peptide microarray with 240 peptides, as described by Hilhorst et al. [20]. Benign and malignant samples were equally distributed over the plates. Each sample was tested in duplicate with ATP and once without ATP on a 96-array plate. For each sample, this test was repeated on a second plate, in an independent experiment. Incubations without ATP were performed to correct for non-specific binding of proteins or antibodies to peptides.

For 13 samples, i.e., 10 PTC and 3 FVPTC (Th-50, Th-52, and Th-53), the effect of kinase inhibitors on STK activity was additionally determined using 0.5 µg of protein and 100 µM of ATP on 96-array plates comprising 140 peptides per array, in the presence of primary antibodies (PamGene International BV, ‘s-Hertogenbosch, The Netherlands), followed by visualization of the signal for 60 min with FITC labeled secondary antibody (PamGene International BV, ‘s-Hertogenbosch, The Netherlands), after washing of the arrays. Incubations were either without inhibitor or in the presence of regorafenib (*n* = 3), sorafenib (*n* = 3) or dabrafenib (*n* = 3). In the experimental design, incubations without and with inhibitor were performed on adjacent arrays on the same strip on 96-array plates (each 96-array plate comprises 24 strips). Final concentrations of regorafenib, sorafenib, and dabrafenib were 50, 10, and 10 µM, respectively. Concentrations were chosen to yield 25% to 50% inhibition of the basal profiles, since these concentrations have proven to be most discriminative. Final DMSO content in the assays was 0.5%.

### 2.6. Data Quantification

After a visual check of all arrays and grids for correct spot finding, signals for each peptide at multiple CCD exposure times were quantified with BioNavigator version 6.3.67 software (PamGene International BV, ‘s-Hertogenbosch, The Netherlands). For each spot on the array, signal intensity after subtraction of local background was used for further analysis. All data processing, statistical analyses, and visualizations were performed using BioNavigator version 6.3.67 software (PamGene International BV, ‘s-Hertogenbosch, The Netherlands).

Signals measured at the end of the incubation were used for analysis. Signals from multiple exposure times were integrated into one value. For the 55 basal profiles, signal intensity was corrected for the signals in the absence of ATP on that 96-array plate and averaged, followed by ^2^log transformation and normalization per 96-array plate to correct for systematic differences between different 96-array plates. After this step, values for replicates obtained on different 96-array plates, were averaged. Only peptides that showed an ATP-dependent signal and an increase in signal intensity in >30% of the arrays with ATP were included in the analysis, resulting in 123 peptides.

For the analysis of the measurements with inhibitors, the same peptide selection was used. Ninety-two of these were present on the 140 peptide arrays used for these experiments. For these samples, an inhibition ratio (log fold change, LFC) was calculated for each peptide by subtracting the ^2^log value without inhibitor from the ^2^log value with inhibitor for the incubations on the same strip. The LFC values of the technical replicates were averaged. The value is equal to zero for peptides for which no changes occur, and <0 for peptides for which inhibition of the relevant kinase activity occurs in the presence of inhibitor.

### 2.7. Classification of Benign and Malignant Samples

Based on the basal profiles of the 55 samples, a class prediction model was built in MATLAB using partial least squares discriminant analysis (PLS-DA) [25] without prior selection of discriminative peptides; coefficients assigned to the peptides reflect the individual contribution of a peptide to the classification model. The performance of the class prediction model was estimated by leave-one-out-cross-validation (LOOCV), ensuring that for each iteration of the cross validation, the model was built independent of the left out sample [26]. Application of this class prediction model to the left out sample results in a prediction score, where prediction score >0 means that the sample is predicted to be benign, and with prediction score <0, the sample is predicted to be malignant. A permutation test was run in which the full 10-fold cross validation procedure was repeated 500 times, but with the ‘benign’ and ‘malignant’ class labels randomly re-assigned to the samples. A detailed description of the method is given in Supplementary Information by Arni et al. [27].

### 2.8. Statistical Analysis

The two-sample two-tailed Student’s *t*-test was used to compare signal intensities per peptide for two group comparisons. The false discovery rate (FDR) was calculated with the Benjamini−Hochberg correction for multiple testing [28]. The statistical analysis was performed with Bionavigator version 6.3.67 software (PamGene International BV).

### 2.9. Upstream Kinase Analysis

In upstream kinase analysis, the phosphorylation changes in peptides on the peptide microarray between two groups are compared and linked to kinases that are known to phosphorylate these sites using several databases and theoretical interactions (PhosphoNet), as described by Chirumamilla et al. [29]. As the peptide sequences on the peptide microarray may be present in multiple proteins, the peptide sequences were blasted in UniProt against human proteins using the PAM30 substitution matrix [30]. Similarity was defined as the ratio of the PAM30 score for a retrieved sequence and the original sequence. Peptides with a similarity score equal to 1 were included. The upstream kinase algorithm provides a list of kinases that might be differentially active between the two groups as described by the kinase statistic, indicating the size and direction of the change, and the kinase score. The ranking of the kinases is based on the kinase score, resulting from the addition of mean significance score and mean specificity score. The mean significance score gives the significance of the change represented by the normalized kinase statistic (direction and size of change in signal) between two groups (using 500 permutations across sample labels), whereas the mean specificity score indicates the specificity of the mean kinase statistic with respect to the number of peptides used for predicting the corresponding kinase (using 500 permutations across target peptides) [29]. The predicted kinases were projected on a phylogenetic tree of the human protein kinase family (courtesy Cell Signaling Technologies, Danvers, MA, USA), using CORAL (http://phanstiel-lab.med.unc.edu/CORAL/ (accessed on 16 January 2023)) [31]. Branch and node color reflect the kinase statistic, node size, and kinase score.

### 2.10. Network Analysis

Protein Atlas (https://www.proteinatlas.org (accessed on 18 January 2023)) was used to check the mRNA and protein presence in both normal and malignant thyroid tissues for the 40 kinases with the highest kinase score. Kinases with a low RNA or protein presence in normal and tumor tissues were excluded, resulting in 29 kinases to build the network. STRING v12.0 (https://string-db.org (accessed on 19 January 2023)) was used to create networks.

## 3. Results

### 3.1. Patient Characteristics

Fifty-five thyroid samples, collected between 2001 and 2010, were included in the study. Histological classifications determined on FFPE sections were confirmed using sections of fresh frozen material. One case (Th-46) judged FVPTC based on the FFPE section, but HP (with Graves’ disease) was based on the fresh frozen section. Since the current work used fresh frozen material, this sample was considered benign. Twenty-one cases were classified as benign, i.e., hyperplasia (HP), follicular adenoma (FA) or oncocytic adenoma (OA), while thirty-four cases were classified as malignant with various histology (Figure 1A).

In summary, the following genetic alterations were found: 

OA (1/3) NRAS Q61R, FTC (1/4) NRAS Q61R, FVPTC (1/3) NRAS Q61R and IDH2 I170T, PTC (8/12) BRAF V600E and ATC (2/10) BRAF V600E, ATC (1/10) HRAS G13R, ATC (1/10) NRAS Q61R, ATC (1/10) IDH1 Q185R, ATC (1/10) fusion HSPA8 (e9)—MET (i3-e4) and ATC (1/10) IDH1 q185r, AQP4 (e1)—RAF1 (i8-e9); FGFR3 (e17-i17)—LCN2 (e7). In three ATC, of which one showed oncocytic characteristics, no genetic alterations were detected. This was also the case for FA (four cases) and OCA (five cases), using targeted sequencing for RAS, RAF, and PIK3CA (Figure 1A).

Seven out of ten lesions with oncocytic characteristics showed extensive whole-chromosome copy number changes, ranging from massive gains (Th-54 and Th-55, OA) to whole-chromosome losses (Th-40, Th-41, Th-42 (OCA), Th-3, and Th-51 (ATC)). Th-35 showed loss of chromosome 22 (Figure 1A). These findings have been reported previously [32]. Relevant clinical and other detailed information, histology, and molecular typing can be found in Appendix A.

### 3.2. Kinase Activity Profiling Classifies Benign and Malignant Thyroid Tumors

Aberrant kinase activity is frequently observed in thyroid tumors, suggesting to us that kinase activity might be able to differentiate aggressive from benign tumors. STK activity was therefore determined in 55 samples, as described in Materials and Methods. HP showed a lower overall STK activity (Figure 1B and Appendix A) compared with various tumor types, and was significantly lower than ATC (*p* = 0.008), PTC (*p* = 0.001), and OCA (*p* = 0.02). The activity found in other benign tumor types (FA, OA) more closely resembled thyroid cancer (Appendix A).

Statistical analysis (Student’s *t*-test) revealed a signal intensity for 82 peptides that was significantly (*p* < 0.05, FDR 8%) different between benign and malignant samples, but a full separation between the benign and malignant groups was not observed using principal component analysis (PCA). While basal signals showed many intra-group commonalities, subtle differences were also observed (see heatmap in Appendix A).

A class prediction model was built using partial least squares discriminant analysis (PLS-DA), while leave-one-out-cross-validation (LOOCV) was used to classify samples and evaluate classification performance.

The model correctly classified 26 of 34 (76%) malignant and 16 of 21 (76%) benign samples (Figure 1C). Of the thirty-four malignant lesions, eight (3/10 ATCs, 2/4 FTCs, 2/5 OCAs, 1/12 PTCs, and 0/3 FVPTCs) were misclassified, with prediction scores ranging between 0.096 and 1.005. Prediction scores for the correctly classified malignant samples ranged from −1.190 to −0.070 (see Appendix A for details). Of the twenty-one benign lesions, five samples (1/4 FA, 2/3 OCA, and 2/14 HP) were misclassified, with prediction scores between −0.103 and −0.442. The correctly classified samples had prediction scores ranging from 0.010 to 1.326.

In addition, 5/10 oncocytic lesions were misclassified, of which four showed extensive chromosomal copy number changes.

The classification accuracy for all samples was 76%, the positive predictive value (PPV) was 72%, and the negative predictive value (NPV) was 87%. A permutation test, where the labels ‘benign’ and ‘malignant’ were repeatedly randomly assigned to the samples, gave a misclassification rate of less than 30% in less than 1% of the permutations. This shows that the obtained classifier has a significant prediction performance (*p* < 0.01).

### 3.3. Molecular Interpretation of Differences between Benign and Malignant Samples

We found that STK activity is increased in malignant tumors (Figure 1B and Appendix A), and using upstream kinase analysis, we identified kinases that are potentially responsible for the differences in peptide phosphorylation between benign and malignant tumors. When the kinases were assigned to a kinome tree, as shown in Figure 2A, an increased activity of members of the ACG kinase family was suggested, which includes cyclic nucleotide binding kinases like PKA, PKG, PRKX, and PRKY, several members of the PKC family, kinases in the mTOR pathway, such as AKT1, AKT2, mTOR, as well as the ribosome-associated kinases, such as p70S6K, RSK, and MSK.

To further investigate these kinases, the presence of the forty kinases with the highest kinase score, i.e., those most likely to be differentially active, was analyzed in both normal and malignant thyroid (www.proteinatlas.org (accessed on 19 January 2023)) (Table 1). After exclusion of the lowest-expressed kinases, a network analysis was performed in STRING for the 29 remaining kinases (Figure 2B). This network contained six interconnected members of the PKC family, five of which link to mTOR. MTOR is the central hub that connects PKCs via AKT and GSK3β to several ribosomal protein kinases (RPS6KA and RPS6KB). PKA and PKG are also connected to GSK3β and the ribosomal protein kinases.

### 3.4. Differentiating Non-BRAF V600E and BRAF V600E PTC Using RAF Inhibitors

We then asked whether ex vivo addition of a BRAF V600E-specific inhibitor to a tumor lysate could be used to distinguish between *BRAF* V600E and non-*BRAF* V600E thyroid tumors, as well as between recurrent and non-recurrent tumors. We compared kinase activity profiles in six non-*BRAF* V600E tumors, including three FVPTC (Th-50, Th-52, and Th-53, annotated as PTC), and seven *BRAF* V600E-mutant PTC (see Appendix A) without and with the *BRAF* V600E-specific inhibitor dabrafenib, and the low-affinity RAF inhibitors sorafenib and regorafenib. Owing to their different histology, the two *BRAF* V600E-positive ATCs were excluded from this experiment.

Basal kinase activity profiles of the *BRAF* V600E samples showed wide variation (median 2log signal intensity 8.27 ± 0.67 for *BRAF* V600E and 8.66 ± 0.35 for non-BRAF V600E tumors, *p* = 0.31) (Figure 3A and Appendix A).

For the 13 PTC samples, the ratio of inhibited to non-inhibited signal (log fold change, LFC) was calculated and represented in heatmaps (Appendix A). Dabrafenib, regorafenib, and sorafenib showed inhibition of overall kinase activity in the tested samples, as demonstrated by altered LFCs of the peptides. However, in a global analysis, median inhibition by regorafenib and sorafenib in *BRAF* V600E and non-BRAF V600E samples was comparable (−0.25 ± 0.1 vs. −0.23 ± 0.08 and −0.27 ± 0.12 vs. −0.31 ± 0.15, respectively), while dabrafenib showed a greater average inhibition of *BRAF* V600E (−0.12 ± 0.2 vs. −0.001 ± 0.15).

(Figure 3B). Compared to regorafenib-treated or sorafenib-treated samples, dabrafenib resulted in more distinct differences between samples (Appendix A).

Statistical analysis of the basal profiles yielded five peptides (ADDB_696-708, ERF_519-531, GSUB_61-73, LMNB1_16-28, and PP2AB_297-309) that showed a significantly different signal (*p* < 0.05, FDR 76%) and was able to differentiate *BRAF* V600E from non-*BRAF* V600E thyroid cancers (Figure 4A). In particular, phosphorylation of the peptide LMNB1_16-28 differed significantly between thyroid cancers (*p* = 0.0006, FDR 6%), and analysis of the same 13 PTC samples from the previous experiment confirmed this finding (LMNB1_16-28: *p* < 0.05, FDR 25%). Since this peptide is a good substrate for several members of the MAPK family, this suggests higher MAPK activity in non-*BRAF* V600E tumors, contrary to the expected increase in MAPK activity in the *BRAF* V600E samples.

Although the biological heterogeneity of *BRAF* V600E samples was substantial, PCA analysis of the basal kinase activity profiles of these PTCs allowed us to distinguish *BRAF* V600E and non-*BRAF* V600E samples (Figure 4B).

Statistical analysis of the LFC profiles with the *BRAF* V600E-specific inhibitor dabrafenib differentiated *BRAF* V600E and non-*BRAF* V600E thyroid cancers, with six peptides (BAD_93- 105, CD27_212-224, MPIP1_172-184, PRKDC_2618-2630, and RB_242-254) showing *p* < 0.05.

(FDR 62%) and one peptide (ANXA1_209_221, Annexin 1) showing *p* < 0.01 (FDR 45%) (Figure 4C). PCA analysis of the log fold change profiles of the PTC exposed to dabrafenib allowed for some distinction of *BRAF* V600E and non-*BRAF* V600E samples (Figure 4D).

In contrast, the low-affinity inhibitors regorafenib and sorafenib showed no significant differences in LFC between *BRAF* V600E and non-*BRAF* V600E PTCs (regorafenib, one peptide with *p* < 0.01, FDR 36%; sorafenib, one peptide with *p* < 0.05, FDR 99%) (Figure 4E and 4F, respectively).

The same data were then used to differentiate recurrent and non-recurrent samples. Using the *BRAF* V600E-specific inhibitor dabrafenib, six peptides (ACM1_421-433, CGHB_109-121, CREB1_126-138, H2B1B_27-40, LMNB1-16-28, and RB_242-254) differed significantly (*p* < 0.05, FDR 47%) between recurrent and non-recurrent samples. Of these six peptides, CGHB_109-121 and LMNB1_16-28 had *p* < 0.01 (FDR 43%). Basal profiles as well as LFC profiles obtained with RAF inhibitors were not able to differentiate these groups: sorafenib, basal, and LFC profiles revealed no significantly different peptides, and only two peptides (CDK7_163-175 and H32_3-18) had a *p* < 0.05 with regorafenib.

## 4. Discussion

### 4.1. Kinase Activity Profiling Correctly Classifies Benign and Malignant Thyroid Tumors

Targeted therapy, which is currently based on NGS, does not always guarantee success and may, in some cases, be accompanied by adverse side effects. Although the mechanisms responsible for failure of targeted therapy are still poorly understood, deregulated kinase activity is thought to play an important role. In many tumor types, deregulated kinase activity has been reported to be associated with DNA variants and tumorigenesis. For example, in non-small cell lung carcinoma (NSCLC), kinase activity profiling of tumor tissue has successfully differentiated short- from long-term survivors [27], and *BRAF* V600E from WT *BRAF* metastatic melanoma [34].

Here, we used serine/threonine (STK) activity profiling to characterize benign and malignant thyroid lesions. Based on our data, a classification model was developed which correctly classified 76% of samples. The misclassification of the remaining 24% of samples, consisting of five benign and eight malignant samples, can be partially explained by the presence of numerous inflammatory cells (Graves’ disease, Th-22 and Th-46, false positive), oncocytic neoplasia (Th-41, Th-42, and Th-51, false negative) associated with reduced kinetic activity, and necrosis (Th-6, Th-47, and Th-44, poor sample quality). Collectively, these data show that increased STK activity is primarily associated with malignancy. As kinase activity profiling can correctly classify 76% of samples, it clearly shows promise as a tool to distinguish benign from malignant tumors in cases where current methods are inconclusive. To more firmly establish the potential of this method, the testing of larger numbers of samples will be needed. Furthermore, exclusion of samples showing necrosis or chromosomal instability will likely improve classification.

### 4.2. Molecular Interpretation of Differences between Benign and Malignant Samples

With a few exceptions, malignant thyroid tumors showed higher STK activity compared with benign lesions. This difference allows us to generate unbiased hypotheses about kinases that are differentially active between benign and malignant samples, irrespective of whether this increased activity is caused by overexpression, chromosomal rearrangements, fusions, mutations of kinases or by factors that modulate kinase activity. This approach could potentially provide new treatment targets for recurrent NMTC. Upstream kinase analysis suggested increased activity of numerous kinases, but only four kinases were associated with thyroid tumors. Mutations in RAF and RAS kinases are most frequent in PTC and FTC, respectively. However, BRAF, ARAF, and RAF1 were not among the top 40 hits in the upstream kinase analysis (ranked 49, 51, and 60, respectively) (see Table 1). Conversely, mutations in higher-ranked kinases such as CHK2 (Table 1, rank 15) or PKD1 (rank 18) are rare (less than 1% of the PTC cases) and do not seem to contribute to thyroid cancer [35]. By comparing the transcriptome of thyroid cancer to the transcriptome of 16 other cancer types, the Protein Atlas shows kinases upregulated in thyroid tumor tissue [36]. Among the 289 genes (of 20,090 human genes) included, three serine/threonine-protein kinases (RSK1, CAMK2B, and STK32A (YANK)) and four tyrosine-protein kinases (ALK, DAPK2, EPHA4, and NTRK2) show elevated expression in thyroid tumor tissue. Of these, only RSK1 was identified in our study (see Table 1). We found an increase in RSK1 (RPS6KA2, rank 39) and PRKX (rank 11) expression, but this was not related to prognosis. The unfavorable prognostic markers PKD1 (rank 18) and PKC (theta) (PRKCQ, rank 17) were not upregulated [37,38]. These observations suggest that modulation of kinase activity, rather than the simple presence of a kinase, is important in malignant tumors. Notably, kinases of the AGC family (i.e., several PKC family members) and PKA, PKG or PRKX, as well as kinases in the AKT/mTOR pathway, show elevated activity in malignant tissue. This same group of kinases was found to have increased activity in vemurafenib-resistant melanoma harboring *BRAF* V600E mutations [34]. Network analysis also pointed to a central role for mTOR/AKT and PKA in malignant samples, similar to the RAS-like PTC subtype as defined by the Cancer Genome Atlas Research Network [35]. A RAS-like subtype was expected, since 25/34 samples were non-*BRAF* V600E. Activation of these hubs occurs through two routes: via cyclic nucleotide second messengers and PIP3. The cyclic nucleotide second messengers, cAMP and cGMP, activate PKA, PKG1, and PKG2. PKA and cAMP seem to protect thyroid cells from apoptosis and support a contribution of p70S6K to viability [39,40,41,42,43,44].

Kinase activity profiling of tumor lysates with multiple drugs may reveal alternative treatment options. Recent work from Huo et al. [45] highlighted the utility of drug screening in PDXs carrying the BRAF G469V variant by identifying an unexpected therapeutic target for EGFR inhibitors. Our analysis (see Table 1 and Figure 2) suggests an important role for members of the PKC family, including the unfavorable prognostic marker PKC (theta) (PRKCQ, rank 17). PKC family members show a strong connection to PKC (epsilon) (PRKCE, rank 3) and PKC (beta) (PRKCB, rank 16). These network hubs, in addition to PKD1 and PKC (theta), merit further investigation and might provide new targets for treatment of NMTC. Inhibitors designed to target these kinases are at several stages of development, including a PKC (theta) inhibitor at the clinical stage [46] and several PKD inhibitors in preclinical development [47].

### 4.3. Distinguishing Non-BRAF V600E from BRAF V600E PTC Using RAF Inhibitors

New treatments are urgently needed for thyroid cancer patients with radioactive iodide refractory disease. While kinase inhibitors such as sorafenib, lenvatinib or cabozantinib can be beneficial, their use is limited owing to significant adverse side effects as well as the development of resistance. Treatment with *BRAF* V600E-specific inhibitors, alone or in combination with an MEK inhibitor, may be a viable alternative. BRAF V600E-targeted mono- and combination therapy is effective in *BRAF* V600E melanoma [17,48], but is ineffective against *BRAF* V600E colorectal cancer [49,50]. Dabrafenib has been approved for ATC thyroid tumors carrying this mutation [51], but not for PTC [15,16,19,52,53,54].

Using STK activity profiling and by spiking with dedicated *BRAF* V600E kinase inhibitor(s), we also investigated whether the effect of RAF inhibitors differs between non-BRAF V600E and *BRAF* V600E PTCs, and between recurrent and non-recurrent tumors. In a study of melanoma samples harboring the *BRAF* V600E mutation, this approach could distinguish responders and non-responders to dabrafenib monotherapy [34].

In our study, basal kinase activity profiles and LFC profiles with dabrafenib could distinguish *BRAF* V600E from non-*BRAF* V600E tumors based on differential phosphorylation. This was not the case for the kinase inhibitors sorafenib and regorafenib (see Figure 4E,F). Dabrafenib could also successfully distinguish refractive from non-refractive tumors. However, since in all cases the variation in profiles between groups was large and the aforementioned differences were due to a small number of peptides with low signal intensities, these results are unlikely to provide a robust biomarker to help differentiate these groups. However, differences in ex vivo inhibition by dabrafenib might reliably predict responders and non-responders to BRAF inhibitor therapy, as shown for melanoma [34]. This could be beneficial for a subgroup of PTC patients, because vemurafenib and dabrafenib have been shown to not only inhibit the RAF/MEK/ERK pathway in thyroid tumors, but also to induce redifferentiation and radioactive iodide restoration [15,16,19,52,53] via re-expression of the sodium/iodide symporter (NIS) [54].

In conclusion, as STK activity profiling allowed us to correctly classify 76% of thyroid tumor samples, this method appears to be able to differentiate benign from malignant thyroid tumors. Moreover, a plausible explanation was found for ten of the thirteen misclassified samples. Molecular interpretation of these differences prompted hypotheses concerning the differential activity of kinases in benign versus malignant tumors. Two of these kinases, PKC (theta) and PKD1, have been previously reported to be prognostic in thyroid cancer [35,55] and might represent new targets for treatment. In addition, ex vivo administration of dabrafenib showed increased inhibition of BRAF V600E in PTC.

We have shown that STK activity profiling can be a beneficial tool for differentiating benign and malignant thyroid tumors. In addition, our approach aids in the selection of (novel) kinase inhibitors for treatment of recurrent thyroid and other cancers.

## Figures and Tables

**Figure 1 cancers-15-04477-f001:**
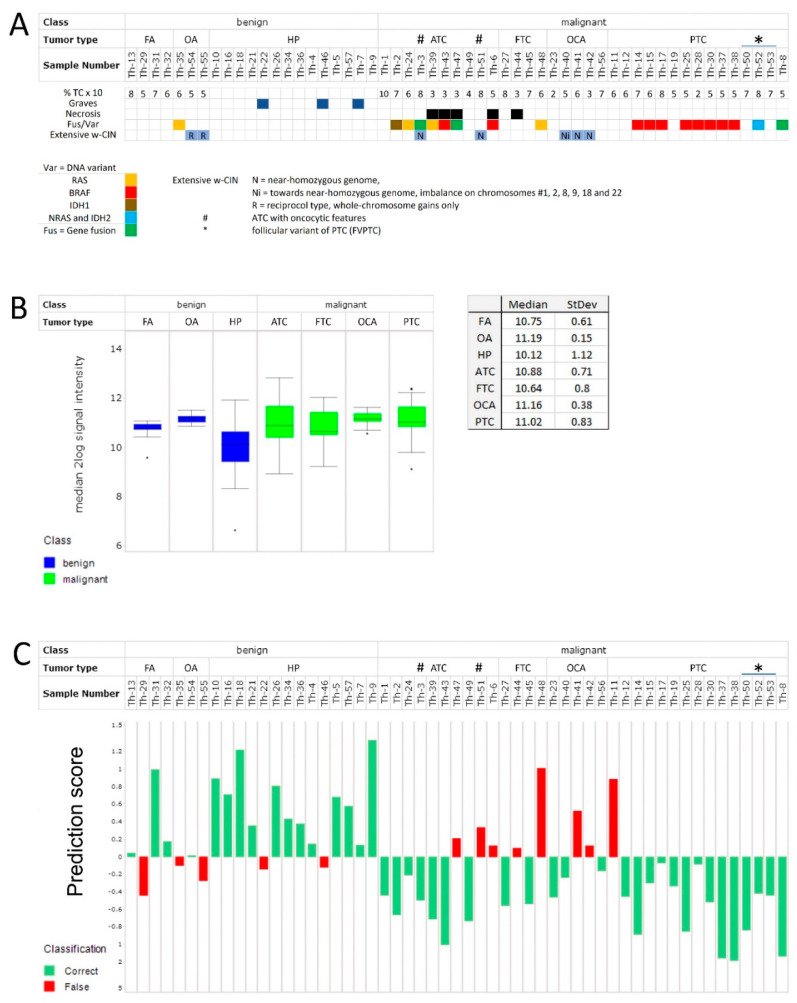
Synopsis of patient cohort and serine/threonine kinase activity profiles and classification of thyroid tumors. (**A**) Benign and malignant thyroid tumor samples of this study were arranged by histological subtype based on the evaluation of formalin-fixed, paraffin-embedded HE sections. Percentage tumor cells (% TC), whether it concerned a patient with Graves’ disease, presence of necrosis, a fusion or DNA variant (Fus/Var), and extensive whole-chromosome instability (w-CIN) were annotated: the near-homozygous genome type (N) or the reciprocal type (R) [32,33]. (**B**) Box plot showing the median ^2^log signal intensity and upper and lower quartiles per sample (median value of 123 peptides), grouped per tumor type. Note the relatively low activity in HP compared to FA, OA, and thyroid cancers (ATC, FTC, OCA, and PTC). The median activity in HP is significantly lower than in ATC, OCA, and PTC (*p*-values 0.008, 0.021, and 0.001, respectively). (**C**) Classification of benign or malignant thyroid tumor samples using a PLS-DA classifier and Leave-One-Out-Cross-Validation. Samples are colored by their classification. The prediction score indicates their classification. A positive value for the prediction score implies a benign tumor, a negative value a malignant tumor. Eight malignant tumors were misclassified as benign, whereas five benign samples were misclassified as malignant. FA: follicular adenoma; HP: hyperplasia; OA: oncocytic adenoma; ATC: anaplastic thyroid carcinoma; FTC: follicular thyroid carcinoma; OCA: oncocytic carcinoma of the thyroid; PTC: papillary thyroid carcinoma. * FVPTC subtype, # Oncocytic features.

**Figure 2 cancers-15-04477-f002:**
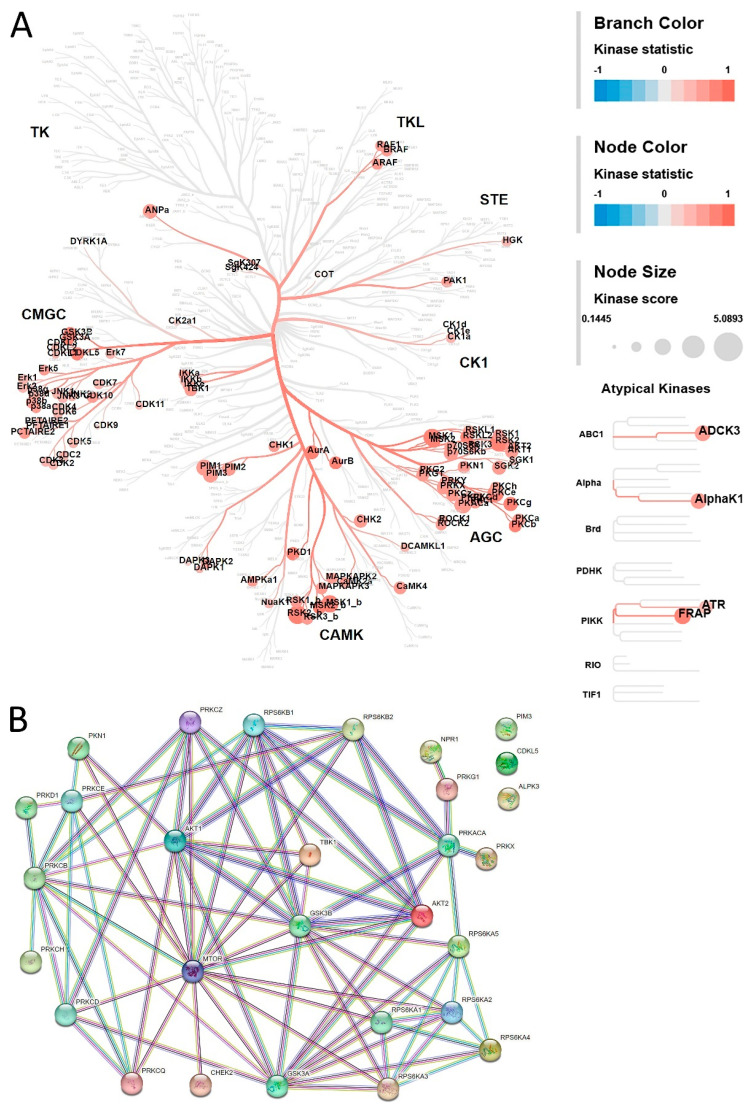
Upstream kinases and STRING Network analysis. (**A**) Results of upstream kinase analysis to identify kinases that may be differentially active between benign and malignant tumors. The kinases were mapped onto a phylogenetic tree of the human kinome using the CORAL application. The branches and nodes are colored by the kinase statistic (i.e., direction and size of the effect). The brighter red the color, the higher the likelihood of a kinase having increased activity in malignant samples. The size of the circle indicates the kinase score of the corresponding kinase (a higher score indicates a higher likelihood to contribute to the observed phosphorylation changes). (**B**) STRING Network analysis of 29 kinases with the highest kinase score for benign vs. malignant tumors (Table 1). Kinases were selected for expression in normal and malignant thyroid tissue using data from Protein Atlas.

**Figure 3 cancers-15-04477-f003:**
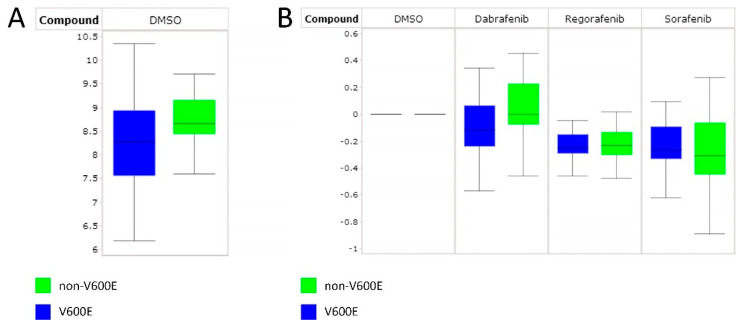
(**A**) Box plot of median ^2^log signal intensity of 13 PTC samples, i.e., seven *BRAF* V600E-positive and six non-*BRAF* V600E (indicated by non-BRAF V600E) thyroid tumor lysates. (**B**) Median log fold changes (log ratio compared to the corresponding untreated sample) of the 13 PTC samples with addition of 10 μM dabrafenib, 50 μM regorafenib or 10 μM sorafenib in the assay. The log fold change for the DMSO controls is equal to 0. Th-50, Th-52, and Th-53 comprise the FVPTC subtype.

**Figure 4 cancers-15-04477-f004:**
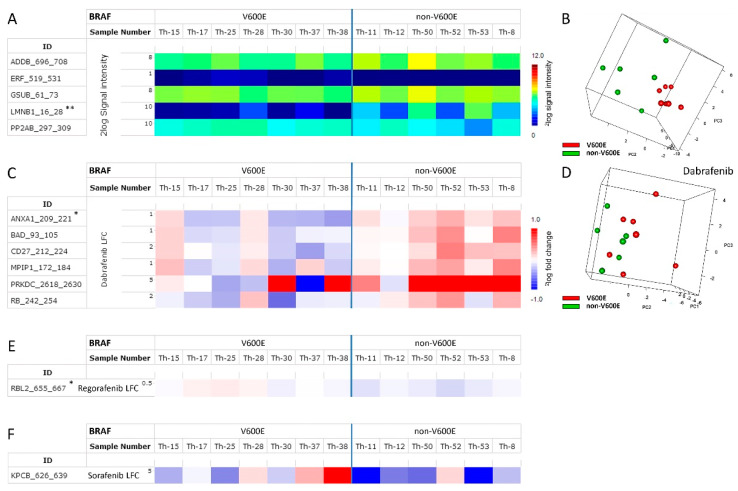
Peptides significantly different (*p* < 0.05) between *BRAF* V600E-positive and non-*BRAF* V600E. PTC samples for basal kinase activity profiles (**A**) or log fold changes (LFC) after ex vivo addition of dabrafenib (**C**), regorafenib (**E**) or sorafenib (**F**). * indicates *p* < 0.01, ** indicates *p* < 0.001. (**B**,**D**) Separation of *BRAF* V600E-positive and non-*BRAF* V600E PCT by basal kinase activity profiles (**B**) or LFC with dabrafenib (**D**) after principal component analysis (PCA).

**Table 1 cancers-15-04477-t001:** Kinases, differentially active in benign vs. malignant tumors, ranked by kinase score after upstream kinase analysis. The presence of RNA and protein in normal and tumor thyroid tissues and the quality assessment of protein presence as checked from Protein Atlas (www.proteinatlas.org (accessed on 19 January 2023)). Kinases are indicated by their UniProt names and ID.

						Thyroid Presence		
Rank	Kinase	Gene Name	UniProt ID	Kinase Score	Normal Tissue	Cancer	Reliability Score *	Remarks
1	RSK2	RPS6KA3	P51812	5.09	yes	yes	approved	
2	MSK1	RPS6KA5	O75582	4.74	yes	yes	approved	
3	PKC[epsilon]	PRKCE	Q02156	4.46	yes	yes	enhanced	
4	PKA[alpha]	PRKACA	P17612	4.66	yes	yes	approved	
5	Pim1	PIM1	P11309	4.38	low	low	uncertain	
6	SGK2	SGK2	Q9HBY8	3.82	low	low	uncertain	
7	ANP[alpha]	NPR1	P16066	3.98	yes	yes	na	Has kinase-like homology domain
8	p70S6K[beta]	RPS6KB2	Q9UBS0	3.90	yes	yes	approved	
9	PKC[gamma]	PRKCG	P05129	3.89	low	low	enhanced	
10	Pim3	PIM3	Q86V86	4.06	yes	yes	na	
11	PRKX	PRKX	P51817	3.96	yes	yes	supported	Thyroid cancer enhanced
12	AurA/Aur2	AURKA	O14965	3.77	low	low	enhanced	
13	Pim2	PIM2	Q9P1W9	3.71	low	low	na	
14	MSK2	RPS6KA4/RSKB	O75676	3.70	yes	yes	approved	
15	CHK2	CHEK2	O96017	3.64	yes	yes	enhanced	
16	GSK3[beta]	GSK3B	P49841	3.66	yes	yes	approved	
17	PKC[theta]	PRKCQ	Q04759	3.62	yes	yes	na	Unfavorable prognostic marker
18	PKD1	PRKD1	Q15139	3.62	yes	yes	uncertain	Unfavorable prognostic marker
19	PKC[delta]	PRKCD	Q05655	3.74	yes	yes	enhanced	
20	Akt1/PKB[alpha]	PKB (alpha)	P31749	3.61	yes	yes	supported	
21	Akt2/PKB[beta]	PKB (beta)	P31751	3.58	yes	yes	uncertain	
22	CDKL5	CDKL5	O76039	3.59	yes	yes	approved	
23	GSK3[alpha]	GSK3A	P49840	3.45	yes	yes	approved	
24	AurB/Aur1	AURKB	Q96GD4	3.49	low	low	enhanced	
25	p70S6K	RPS6KB1	P23443	3.45	yes	yes	approved	
26	RSKL2	RPS6KL1	Q9Y6S9	3.49	low	low	approved	
27	TBK1	TBK1	Q9UHD2	3.36	yes	yes	uncertain	
28	PKC[alpha]	PRKCA	P17252	3.47	low	low	enhanced	
29	PKC[beta]	PRKCB	P05771	3.46	yes	yes	approved	
30	CaMK4	CAMK4	Q16566	3.49	low	low	approved	
31	PRKY	PRKY	O43930	3.28	na	na	na	
32	PKC[zeta]	PRKCZ	Q05513	3.40	yes	yes	enhanced	
33	ADCK3	COQ8A	Q8NI60	3.27	na	na	na	
34	PKG1	PRKG1	Q13976	3.48	yes	yes	enhanced	
35	PKN1/PRK1	PKN1	Q16512	3.38	yes	yes	approved	
36	PKC[eta]	PRKCH	P24723	3.30	yes	yes	approved	
37	mTOR/FRAP	FRAP	P42345	3.31	yes	yes	approved	
38	RSK3	RPS6KA1	Q15418	3.30	yes	yes	approved	
39	RSK1/p90RSK	RPS6KA2	Q15349	3.27	yes	yes	enhanced	Thyroid cancer enhanced
40	AlphaK1	ALPK3	Q96L96	3.25	low	yes	uncertain	

* Reliability score for protein presence in normal tissue, based on knowledge-based evaluation of available RNA-seq data, protein/gene characterization data, and immunohistochemical data from one or several antibodies designed towards non-overlapping sequences of the same gene. The reliability score is based on the 44 normal tissues analyzed (https://www.proteinatlas.org/about/assays+annotation#ih_reliability (accessed on 20 January 2023)). Reliability score from high to low: Enhanced, Supported, Approved, Uncertain; na: evidence based on RNA expression, no evidence for protein presence reported.

## Data Availability

The data can be shared up on request.

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
