# Peer review of "Differentiating Benign from Malignant Thyroid Tumors by Kinase Activity Profiling and Dabrafenib BRAF V600E Targeting"

_cancers, 2023, doi:10.3390/cancers15184477_

Round 1
Reviewer 1 Report
Hilhorst et al present a very fresh report on kinase activity profiling in radioactive iodine-resistant thyroid cancer. I think that the methodology applied by them is very promising and worthy of scientific interest. The analysis is sound and the results are intriguing. I approve the manuscript as it is
Author Response
Thank you for your review, positive comments and approval of our work. This is highly appreciated.
Reviewer 2 Report
The paper is somehow interesting, but in light of the scientific literature nowadays available does not provide a great enahncement of knowledge. Moreover the presumed differentiation potential of using specific inhibitors in a clinical setting is really overcome by the next generation sequencing technologies that can detect with high confidence the presence of revelant mutations in the tumors..Therefore I do not think this article is innovative enough for this journal
Author Response
As this reviewer remarks, whole-genome sequencing (NGS) has opened many new opportunities for targeted treatment with e.g. kinase inhibitors. For several cancers, the mutations found with NGS have been used in so called “basket trials” to guide targeted treatment. However, these trials show a large discrepancy between the presence of a certain mutation and the response of the tumor to the specific inhibitor. Understanding tumor response targeting inhibitors requires integration of knowledge of all mutations in such a tumor: e.g. gene expression, translation and post-translational modifications. So knowledge about a certain mutation, e.g. BRAF V600E, does not certify a treatment response.
In this study we do not demonstrate the presence of proteins, as achieved with e.g. mass spec studies, but determine the activity of all kinases, decorated with the relevant post-translational modifications. The large variations in kinase activity (for tumors having all the same mutation) demonstrate that the
physiology is very complex. The integrated view of kinase activity might provide a better prediction of response to treatment. Several publications [1-7] show that the effect of mutations in relation to response to treatment can be determined with kinase activity profiling, in the presence or absence of a specific inhibitor is tested.
Therefore we think that kinase activity profiling is a technology that supplements other technical platforms, like NGS.
With the comments of the reviewer in mind we have critically read the manuscript and adjusted several sections to strengthen the link between results and conclusions. All changes in the text were yellow high- lighted, in order to facilitate tracking.
References (Reply reviewer 2)
- Ruijtenbeek, R., et al., Differential protein kinase activity in ER-positive and ER-negative breast cancer. Journal of clinical oncology : official journal of the American Society of Clinical Oncology, 2009. 27(15_suppl): p. e22142.
- Hilhorst, R., et al., Peptide microarrays for detailed, high-throughput substrate identification, kinetic characterization, and inhibition studies on protein kinase Analytical biochemistry, 2009. 387(2): p. 150-61.
- Hilhorst, , et al., Blind prediction of response to erlotinib in early-stage non-small cell lung cancer (NSCLC) in a neoadjuvant setting based on kinase activity profiles. Journal of clinical oncology : official journal of the American Society of Clinical Oncology, 2011. 29(15_suppl): p. 10521.
- Hilhorst, R., et al., Peptide microarrays for profiling of serine/threonine kinase activity of recombinant kinases and lysates of cells and tissue samples. Methods in molecular biology (Clifton, N J ), 2013. 977: p. 259-71.
- Hilhorst, R., et al., Peptide microarrays for profiling of serine/threonine kinase activity of recombinant kinases and lysates of cells and tissue Methods Mol Biol, 2013. 977: p. 259- 71.
- Krayem, , et al., Kinome Profiling to Predict Sensitivity to MAPK Inhibition in Melanoma and to Provide New Insights into Intrinsic and Acquired Mechanism of Resistance. Cancers (Basel), 2020. 12(2).
- Roy, , et al., Identification of Novel Substrates for cGMP Dependent Protein Kinase (PKG) through Kinase Activity Profiling to Understand Its Putative Role in Inherited Retinal Degeneration. Int J Mol Sci, 2021. 22(3).
Reviewer 3 Report
1. The quality of the figures is low please improve them.
2. The abstract does not provide enough context or background information about the study. And also it needs more clarity.
3. The conclusion should have more information about the subject and results.
4. To show the chemical structure of some anti-angiogenesis receptor tyrosine kinase inhibitors, especially lenvatinib please refer to this review article after "Kinase inhibitors, such as sorafenib, lenvatinib or cabozantinib can 504 be beneficial."
Minor editing of English language ia required.
Author Response
We wish to thank this reviewer for helpful comments.
- “The quality of the figures is low please improve ”
We agree that the quality of the figures is somewhat disappointing after downloading the figures for reviewing. However, we can guarantee the reviewer that all figures were designed at 600 dpi making use of PhotoShop. Apparently, the resolution is down-scaled after uploading the files to the MDPI-server. The final quality of the figures is in the hands of the publisher.
- “The abstract does not provide enough context or background information about the And also it needs more clarity.”
- “The conclusion should have more information about the subject and ”
We made substantial rewriting of the abstract and the manuscript. Furthermore, we send out the manuscript to an editing service (https://www.medicalediting.nl). Several sections of the manuscript, including abstract and conclusions have been revised to improve clarity. All changes were yellow high- lighted.
- “To show the chemical structure of some anti-angiogenesis receptor tyrosine kinase inhibitors, especially lenvatinib please refer to this review article after "Kinase inhibitors, such as sorafenib, lenvatinib or cabozantinib can 504 be beneficial."”
We discussed the addition of the compounds chemical formula’s as a Supplementary Figure S3. However, the chemical structures of the compounds can be easily found on the internet, e.g. Wikipedia. We therefor share the opinion that this is a bit superfluous and left the chemical structures of the used drugs out of the manuscript.
Reviewer 3 points to a review article of which he or she would like to be seen cited and we recite: “please refer to this review article after …” However, the reviewer left out to specify this review paper. Consequently we did not cite this paper.
Round 2
Reviewer 2 Report
The Autor greatly improved the clarity and readability of the manuscript. No additional requests to be integrated in the text that should be published in the present form